# Classification of glucose-level in deionized water using machine learning models and data pre-processing technique

Tri Ngo Quang[1,2], Tung Nguyen Thanh[1,3]*, Duc Le Anh[1], Huong Pham Thi Viet[1], Doanh Sai Cong[4]

1 International School, Vietnam National University, Hanoi, Vietnam, 2 Faculty of Information Technology, University of Economics-Technology for Industries, Hanoi, Vietnam, 3 Faculty of IT, Nguyen Tat Thanh University, Ho Chi Minh City, Vietnam, 4 University of Science, Vietnam National University, Hanoi, Vietnam

* tung_nt@vnu.edu.vn

## Abstract

Accurate monitoring of glucose levels is essential in the field of diabetes detection and prevention to ensure appropriate treatment planning. Conventional blood glucose monitoring methods, although widely used, are intrusive and frequently result in discomfort. This study investigates the use of Raman spectroscopy as a non-invasive method for estimating glucose concentrations. Our proposition entails employing machine learning models to categorize glucose levels by utilizing Raman spectrum data. The collection consists of deionized water samples containing glucose with defined amounts, guaranteeing great purity and little interference. We assess the efficacy of three machine learning models in categorizing glucose levels which including Extra Trees, Random Forest, and Support Vector Machine (SVM). In addition, we employ data pre-processing techniques such as fluorescence background removal and hotspot series extraction to improve the performance of the model. The primary results demonstrate that the utilization of these pre-processing techniques greatly enhances the accuracy of classification. Among these techniques, the Extra Trees model achieves the highest accuracy, reaching 95%. This study showcases the viability of employing machine learning techniques to forecast glucose levels based on Raman spectroscopy data. Additionally, it emphasizes the significance of data pre-processing in enhancing the accuracy of the model's results.

**Data Availability Statement:** The datasets are generated and stored by the University of Science, Viet Nam National University, Hanoi. The datasets during the current study are not publicly available

## 1. Introduction

The detection and management of diabetes hinge critically on the accurate monitoring of blood glucose levels. Traditional methods, such as blood glucose testing through pinpricks or implanted lancets, are invasive and often uncomfortable for patients [1]. Specifically, in the case of Glucose Tolerance Test (GTT) [2], patients must experience a modest number of glucose tests during a test. To address these challenges, non-invasive techniques for glucose monitoring have been actively explored, with Raman spectroscopy emerging as a promising approach. In paper [3], several drawbacks of invasive glucose tests are indicated and non-

due to Viet Nam National University but are available from the University of Science on reasonable request via email at hus@vnu.edu.vn Alternatively, requests for dataset availability can be sent to the corresponding author, Nguyen Thanh Tung, at email: tung_nt@vnu.edu.vn.

**Funding:** This research is funded by Ministry of Science and Technology (MOST) under project number DTDL.CN-25/23.

**Competing interests:** The authors have declared that no competing interests exist.

invasive glucose monitoring based on sensors is indicated.Raman spectroscopy leverages the scattering of monochromatic light to measure the molecular composition of biological samples, offering a non-invasive means to estimate glucose concentrations. Despite its potential, non-invasive glucose monitoring using Raman spectroscopy faces significant obstacles, including the difficulty of data collection and the challenge of achieving high accuracy in glucose level estimation [4]. Previous studies have demonstrated the feasibility of this approach, yet consistent and reliable prediction remains elusive due to the complex nature of Raman spectral data and the noise inherent in non-invasive measurements.

In this study, we build upon prior research by integrating machine learning techniques with Raman spectroscopy data to classify glucose levels. Our work proposes a novel method for predicting glucose concentrations using machine learning models, specifically Extra Trees, Random Forest, and Support Vector Machine (SVM). We focus on two key objectives: validating the potential of machine learning for glucose level prediction and evaluating the impact of various data pre-processing techniques on model performance. To achieve these goals, we conducted experiments using deionized water samples with known glucose concentrations, ensuring data purity and reducing noise. We employed advanced data pre-processing methods, including fluorescence background removal and hotspot series extraction, to enhance the quality of the input data for machine learning models. Our results indicate that these pre-processing techniques significantly improve the accuracy of glucose level classification.

This paper is structured as follows: we first discuss the theoretical background of non-invasive glucose measurement using Raman spectroscopy and the application of machine learning in this domain. We then detail our experimental setup, including data collection and pre-processing methods, followed by a comprehensive evaluation of our machine learning models. Finally, we present our findings, discuss their implications, and outline future research directions. Thus, the paper consists of four major sections: theoretical background, technique for analyzing and pre-processing Raman datasets, experiments, and conclusion.

Through this study, we demonstrate the feasibility of using machine learning to predict glucose levels from Raman spectral data and highlight the importance of effective data pre-processing. Our contributions provide a foundation for future advancements in non-invasive glucose monitoring technologies, potentially improving the quality of life for individuals with diabetes.

## 2. Related works and description of methodology

In this chapter, we describe some general theoretical findings about our project, including non-invasive glucose-level measurement using the Raman spectroscopy and machine learning.

### 2.1. Non invasive—glucose measurement using Raman spectroscopy

Raman measurement used the Raman effect, which was discovered by Raman and Krishnan in 1928 [5]. Raman spectroscopy, based on this effect, is a scattering technique. When a sample is exposed to monochromatic laser sources, the molecules in the sample interact with the lasers and scatter light. A Raman spectra is produced by scattering light at a frequency different from inelastic scattering. The inelastic interaction of sample molecules with monochromatic light generates the Raman spectrum. The measuring instrument that uses Raman spectroscopy is known as a Raman spectrometer and consists of four modules: a laser generator, a chamber for storing measuring samples, a grating-equipped spectrometer chamber, and a detecting system [6].

Before the use of Raman spectroscopy in healthcare was found to be effective, the measurement acquired by non-invasive methods was of concern. For example, Caduff et al. proposed a non-invasive method to predict glucose level from impedance spectroscopy [7]. With the relationship between Raman data and glucose-level, there were several studies that raised huge concerns in the scientific community. Xu et al. [8] described available targets of measurement using Raman spectroscopy as a large amount of substances. Thus, the relationship between Raman spectroscopy reflected from human bodies and blood glucose-level was discussed in [9]. According to this study, blood glucose levels in living animal bodies can be determined in vivo using the Raman spectra obtained from a diode-laser operating at 785 nm. By using partial least squares techniques, the glucose-level of humans can be measured using near-infrared Raman spectroscopy [10]. The research in [11] had a similar approach, but the statistical techniques were classical least squares, principal component regression and partial least squares.

These investigations demonstrate that it is entirely feasible to forecast the glucose level using Raman spectroscopy. Due to the significant difficulty in observing the relationship between Raman data and glucose level, the problem we had to solve was the selection of a prediction technique. We may also notice that several statistical methods have demonstrated their effectiveness in identifying characteristics between Raman spectroscopy and glucose levels. According to previous research and our own findings, the machine learning technique is just as effective as the methods mentioned above. The possibility and approach of machine learning algorithms for glucose prediction based on Raman spectroscopy were addressed in the following section.

## 2.2. The use of machine learning for predict glucose-level from Raman spectra

**2.2.1. The potential of machine learning in glucose concentration prediction.** Machine learning has proved its efficiency in problems of prediction and classification [12]. There were several former studies about the application of machine learning on glucose-level classification from extracting Raman data. Some of them concentrated on detecting diabetes from this kind of data. The research in [13] proposed a non-invasive glucose measurement using 5 simple machine learning algorithms from optical sensors. In the study, the dataset was the 18 pairs which each pair had 2 values: wavelength and intensity. The techniques for machine learning included a Feedforward Neural Network for multi-class classification. In this investigation, we investigated the experimental methodology employed by the authors when they collected glucose-distilled water samples to simulate human blood. The objective of this method is to generate a simple dataset with a high degree of accuracy and a predicted value.

In article [14], the author collected Raman spectroscopy data on blood samples from normal people and diabetes patients before proposing an algorithm based on a combination between Principal Component Analysis and Linear Discriminant Analysis to classify [14]. The advantage of the study, when compared to [13] is that Raman spectroscopy was the sole input data to be classified by the algorithms. As reported by [15], the results and classifcation discretization framework of an experiment using a visible near-infrared laser derived from an optical sensor to measure glucose level were presented and shown to be promising.However, Raman spectroscopy was performed on human blood, which had a high degree of accuracy despite the intrusive nature of blood collection. In addition, 76 individuals were sampled, including 39 diabetes patients and 37 healthy individuals. The small quantity of this number reduces its persuasiveness. The research in [16] introduced a portable spectrometer for non-invasive glucose testing based on the Raman effect. Positive aspect of the study is the in vivo studies conducted on diabetes patients and healthy individuals. Using the spectrometer, the solution acquired

Raman spectroscopy from certain regions of the human body via a non-invasive measurement. Support Vector Machine and Artificial Neural Network were used as machine learning prediction models. Some other benefits of the study were the data processing techniques that enhanced the performance of machine learning algorithms. The first problem with this study is the small sample size, which consists of just about 20 individuals. The second categorization is lean, for which it is hard to specify a precise glucose level. Instead, the answer predicted whether or not a sample was obtained from a diabetic patient.

We acquired a number of outstanding perspectives and approaches from the aforementioned studies, while avoiding their flaws. We continue to reference these research while focusing on the outcome and measuring metric. In this section, we present simply the approach and theoretical foundation.

**2.2.2. Applied machine learning model.** In the scope of study, we use 3 machine learning models for classification, including Extra Trees, Random Forest and Support Vector Machine (SVM):

• The Extra Trees model:

The Extra Trees machine learning model was proposed based on tree architecture [17]. The Extra-Trees approach follows the traditional top-down method and employs a collection of unpruned decision or regression trees. The number of characteristics that are randomly chosen at each node and the minimal sample size for splitting a node are the two parameters for the Extra-Trees splitting technique for numerical attributes. It is applied several times to the whole original learning sample in order to produce an ensemble model, which we identify by the number of trees in the ensemble. By majority vote in classification problems and arithmetic average in regression problems, the predictions of the trees are combined to get the final prediction.

The Extra Trees model has many applications in detection and classification. A solution for detecting phishing websites integrated Extra Trees algorithms and Artificial Intelligence Meta-Learners techniques have been proposed in [18]. Another application of Extra Trees is the Autotrophic/Heterotrophic Microorganism Mixtures using absorbance spectrum data [19].

• The Random Forest model:

The Random Forest model was created by integrating many tree-based data structures and randomizing the dataset and was found to be effective classifier for bio-sensor data [20]. A number of tree classifiers were integrated with Random Forest based on combination regulation. Each of these classifiers casts a vote for the class with the most members, and the final sort result is produced by combining these votes. This algorithm is distinguished by its high classification precision, tolerance for noise and outliers, and lack of overfitting. During the creation of Random Forest, the tree is also planted on the new training set using random feature selection, and the new training set is taken from the previous training set using bagging methods.

Similar to other methods for machine learning, Random Forest can be used for classification and regression. In [21], several uses of Random Forest for managing complex data from remote sensors are discussed. From the list of these applications, we determined that Random Forest was capable of dealing with a number of data sources, such as multi-spectral radar, sensing images, and hyperspectral imagery. Using Raman spectroscopic data and Random Forest has another use: locating substances used in nanofabrication [22]. The computationally feasible analysis of genome-wide association data using Random Forest is one of the first instances presented in this work.

- The Support Vector Machine model:

The Support Vector Machine (SVM) has a huge variety of supervised applications with different data sources [23]. For example, several applications of SVM in the hydrology industries were described in paper [24]. The capacity of the SVM to learn data classification patterns with a balance between accuracy and reproducibility is what gives it its power. It has gained popularity as a classification tool, though it is still infrequently employed for regression tasks. It is highly versatile and may be utilized in a variety of data science contexts, including the study of brain illnesses. In order for the SVM to function, a hyperplane that optimizes the separation between the support vectors of the two class labels was selected. In comparison to other kinds of classifiers, the SVM's capability and attraction stem mostly from its ability to give balanced performance even when the complexity of the feature space greatly exceeds the number of training data. In addition, the SVM provides diversity. For the SVM decision functions, many distinct kernel functions can be provided, and most software enables users to choose unique kernels. This functionality makes it easier to employ the SVM classifiers to solve linear classification problems without having to spend a lot of time on hyperparameter adjustment. The SVM is effective in solving a variety of classification issues with high dimensional data [25]. Therefore, it can be applied efficiently to Raman spectra datasets. Both three machine learning models, including Extra Trees, Random Forest and SVM were fully supported in Sci-kit Learn–a Python library for Machine Learning implementation. Thus, we use this for developing our project.

## 2.3. Description of methodology

The research team is focused on three main aspects: data collection, the proposal of artificial intelligence algorithms, and data preprocessing methods.

Given the unique nature of this study, which aims to demonstrate the capability of classifying Raman spectral sequences from liquid samples with varying glucose concentrations, the data collection will be conducted by the research team rather than relying on existing datasets. This allows the data characteristics to be tailored and the research to be enriched more effectively. For data preprocessing methods, we also refer to data preprocessing techniques related to time series data as well as Raman spectroscopy, thereby constructing a solution that has proven effective in previous studies. Additionally, we represent the collected data, analyze it, and propose effective data preprocessing methods.

For the proposed machine learning models and data preprocessing, the machine learning models' task is to classify the Raman sequences based on the glucose concentration in the sugar water samples they reflect. Meanwhile, the role of the data preprocessing algorithms is to enhance classification performance. Various artificial intelligence algorithms and data preprocessing techniques will be explored, with related works being studied from reputable databases like IEEE Xplore or Springer, as well as conducting an analysis of the collected data. Based on this research, several machine learning algorithms and data preprocessing methods will be proposed and implemented, and their performance will be evaluated through experiments with the collected data. The most effective artificial intelligence algorithms for processing sequential data, such as Raman spectroscopy, are deep learning models such as Convolutional Neural Networks or Long Short-Term Memory Networks. The advantage of deep learning models in data processing lies in their excellent capability to handle complex data, particularly when dealing with noisy data. However, the drawbacks of deep learning models include their high complexity, long training times, and a tendency to overfit when the sample size is too small. Given the current dataset, the use of state-of-the-art technologies like deep learning models is not necessary because the samples clearly exhibit the characteristics of the labels, as

they were collected directly from pure glucose solutions and have a very low noise ratio. More-over, with the limited number of samples, employing deep learning models with strong feature extraction capabilities could lead to overfitting. In this paper, the research team strategically uses data preprocessing techniques to enhance the effectiveness of artificial intelligence models, rather than applying overly powerful models. Moreover, we use fine-tuning mechanisms to set the best value of all each parameter in each machine learning models.

## 3. Introduction to the dataset

### 3.1. Collection and description to the dataset

To evaluate the accuracy of machine learning models in predicting glucose levels for the objectives of this study, sample-level precision was crucial. According to prior study, there are three drawbacks to the samples collected through non-invasive or invasive measures in regions of the human body [13]. The dense appearance of noise, the difficulty of obtaining a sample with the desired value, and the medical ethics surrounding human experimentation. We produce a pseudo-sample with the desired value based on these challenges.

With the purpose of replicating blood samples with a determined glucose level in our laboratory, we created samples by combining pure glucose and deionized water in a particular ratio. Main components used for measurements consist of:

- Raman spectrophotometer: uRaman—Ci, Technospex.

- Glucose chemical products: 99.5% Sigma-Aldrich glucose.

- Deionized water solution bottle.

- Tools (solution tube, stirring rod,...).

The measurement procedure is described as below:

1. Set excitation laser power to 100mW.

2. Set the measurement range at the wavelength of 785.1nm in 300s.

3. Place 3 ml of glucose solution from each of the mentioned test tubes on the quartz surface of the Cuvette.

4. Insert the cuvette (with the MACRO-CH/Quartz Cuvette accessory) into the measurement chamber of the uRaman–Ci spectrophotometer.

5. Collect digitized Raman signals from the measurement experiment.

The set of devices used to acquire Raman spectroscopic data by analyzing samples of glucose-mixed fluids with Technospex uRaman—Ci. We have not yet adopted non-invasive measurement in human tissue due to the side effects of external variables in the Raman data from this measurement were too significant to permit an evaluation of the correlation between glucose level and sample. The Fig 1 is the photo of the Raman spectrometer located in Vietnam National University, which was taken in May 2023.

Within the scope of the study, we collected 5 Raman sequences for each glucose fluid with specific concentration. Therefore, our dataset contains 50 samples with 50 labels in range of 10 values. The Raman shift data is contained in a CSV file, and its label is the prefix of this CSV file, which is separated from the other portions by a "-" character. Each label in the set indicates a glucose value and was encoded as an integer: I = {1, 2, 3, 4, 5, 6, 7, 8, 9, 10}. The encoding process was handled by a function which analyzed the CSV file. Basic description about our dataset is described in Table 1.

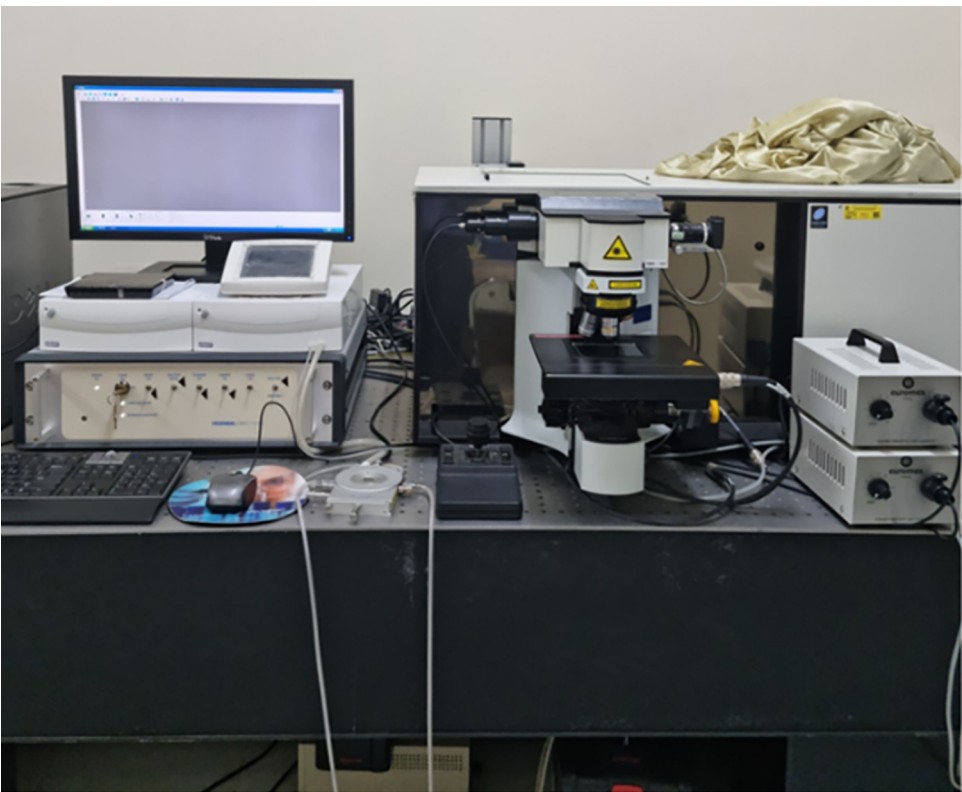

**Fig 1. Device set used for Raman spectroscopy measurement from artificial glucose-mixed fluids.**

The gap between glucose levels is not considered as an extraction criterion because the machine learning technique was built to handle only classification problems. Yet, under conventional settings for measuring blood samples, glucose concentration is the most influential element in determining the result pattern. If the difference in glucose levels between two samples was modest, there would be a high degree of data pattern similarity, which would make machine learning algorithm extraction more challenging. Consequently, the dataset enables us to test the performance of several machine learning models for extracting features from different glucose feature distances.

**Table 1. Description of the dataset used for the machine learning algorithm.**

| Glucose concentration level (mmol/L) | Number of samples | Label code |
| --- | --- | --- |
| 5.0 | 5 | 1 |
| 5.5 | 5 | 2 |
| 6.0 | 5 | 3 |
| 6.5 | 5 | 4 |
| 7.0 | 5 | 5 |
| 7.5 | 5 | 6 |
| 10.0 | 5 | 7 |
| 12.5 | 5 | 8 |
| 15.0 | 5 | 9 |
| 20.0 | 5 | 10 |

### 3.2. The primitive input data

We identify primitive data as the Raman-shift data that was collected directly from a Raman machine with basic noise reduction. Simple noise reduction guarantees that data adhere to the stringent basic rule of light wave shift, which requires that the value be greater than zero. In our dataset, there are 2048 waveshift-intensity pairings for all samples' raw data. Each pair has its own index and is ordered in ascending order by waveshift. Due to the fact that the list of waveshift for each sample is identical, just the intensity and its index are treated as input data for machine learning algorithms.

Based on this description, we describe the input data of each sample as an array of intensity values whose index for each intensity value corresponds to its index in the sample. Throughout the scope of the study, the basic input data for the machine learning method is defined as a 2048-element array of intensity values sorted ascendingly by matching waveshift

## 4. Data preprocessing methods in proposal

### 4.1. The hotspot series extraction procedure

We recognize that the primitive input data of a sample is extremely complicated and needs to be reduced.

**4.1.1. Hotspot segments of primitive data.** In each labeled-group, we randomly selected one of five samples and plotted it using the Python tool matplotlib, as shown in Figs 2 and 3.

Figs 2 and 3 contains two plots with distinct glucose level separations. In Fig 2, the distance is 5, and the range of glucose levels is from 5 to 20. In contrast, the distance in Fig 3 is only 0.5, and the range of glucose levels is between 5.5 and 7.5. We can observe that:

- In all samples, the ascending sequence of wave shift does not provide a persistent change in intensity. There are a considerable number of 10-value curves with the maximums in the initial 1250 indexes. Throughout the last 750 indices, the number has fluctuated modestly.

- In the first indexes of these graphs, the difference between samples' glucose concentrations is expressed clearly. In the range of indexes from 500 to 1250, visibility is excellent.

This finding leads us to the conclusion that the length of input data is enormous, but not all values provide useful classification information for machine learning models. Hence, we

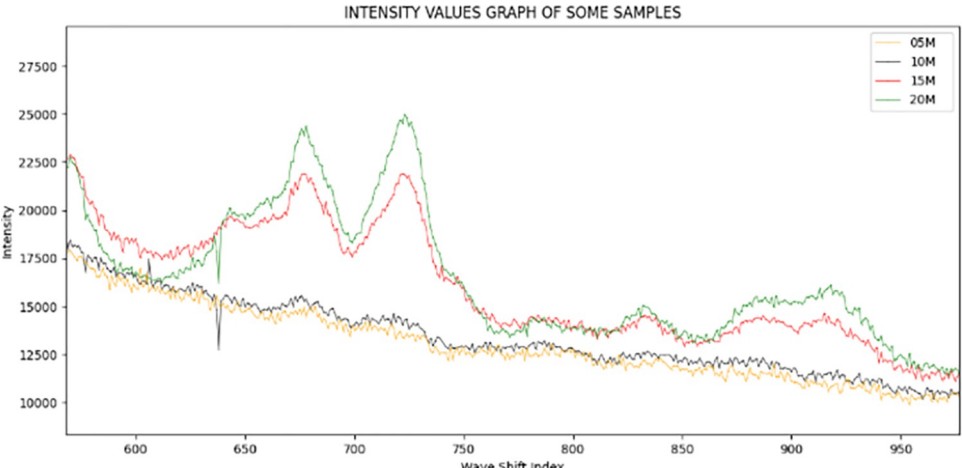

**Fig 2. The graph of intensity values in some samples with different glucose level with large distance between glucose levels of samples: 5, 10, 15 and 20.**

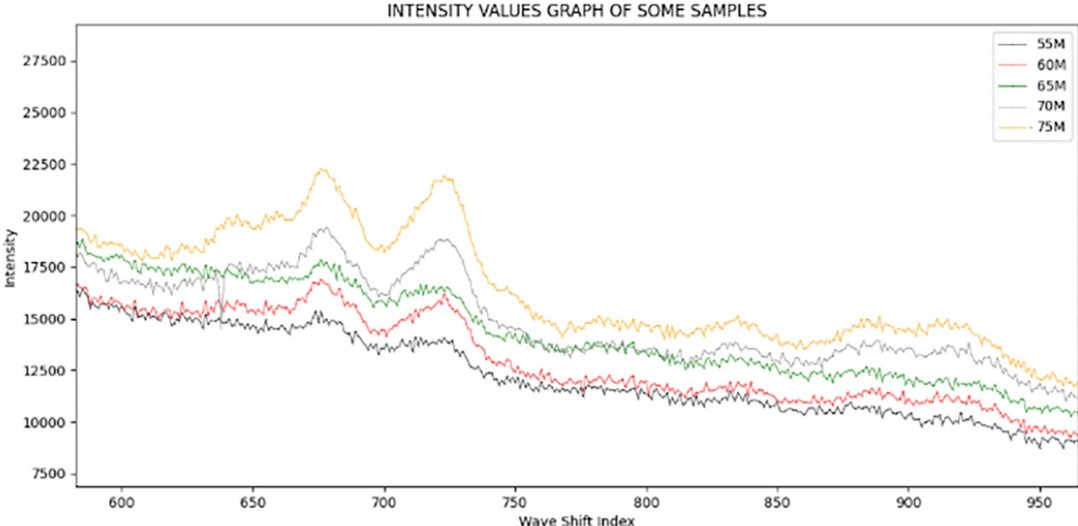

**Fig 3. The graph of intensity values in some samples with different glucose level with small distance between glucose levels of samples: 5.5, 6.0, 6.5, 7.0 and 7.5.**

define a hotspot as a chunk of input data containing useful categorization features. By examining the values of the smaller-than-input-data hotspot region, machine learning models may collect nearly all the characteristics of a sample with a certain glucose level. Although we can define hotspot segments through observation, we build a method to detect them with greater precision.

**4.1.2. Generation of intensity series.** First, we define an intensity series as a portion of input data containing a predetermined number of sequence values. As mentioned previously, each sample's input data is an intensity array. This array can be represented in the following mathematical Formula (1):

$$A_i(g) = \{I_j \in N | j \in *0, 2047]\} \tag{1}$$

In the Formula (1):

- $A_i(g)$: array of intensity values with index i and glucose level g

- $I_j$: intensity value with index j

This array is used to construct an intensity series of length l (l < 2048) by selecting l sequence elements. Hence, the distinct series are produced by assigning distinct elements to the series' initial constituents. The generic Formula (2) provides the mathematical structure of a length l intensity series:

$$S_j(A_i(g)) = \{I_k \in N | l \in *1, 2047],$$

$$j \in [0, 2048 - l], k \in [j, j + l]\} \tag{2}$$

In the Formula (2):

- j: starting index of an intensity series

- $A_i(g)$: array of intensity values with index i and glucose level g

- $S_j(A_i(g))$: an intensity series of $A_i(g)$ with starting index j

- $I_k$: intensity value with index k. This index is setted in $A_i(g)$.

- l: the size of the intensity series

Because the intensity array has 2048 indexes, each sample has a set of (2049—l) series with a separate starting index. From this set, we select the hotspot series and generate a new array that contains all elements of the hotspot series.

**4.1.3. Extraction of hotspot series.** After extracting the intensity series, the next step is selecting the hotspot series and defining the new input data with similar extraction of all samples. Thus, the extraction process includes 3 stages: calculate hotspot level, select the hotspot series based on the hotspot level and output new input data for the machine learning models.

In the first stage, we calculate hotspot level based on median and average variance of this series in the Formulas (3) and (4):

$$S\_j(A_i(g)) = \frac{1}{l} \sum_{j+l}^{j} \left( I_j \right) \tag{3}$$

In the Formula (3):

- $A_i(g)$: array of intensity values with index i and glucose level g

- $\underline{S}_j(A_i(g))$: a median of the intensity series of $A_i(g)$ with starting index j

- $I_j$: intensity value with index j

- l: the size of the intensity series

These is Formula (4)

$$V_j(A_i(g)) = \frac{1}{l} \sum_{j+l}^{j} \left( I_j - S\_j(A_i(g)) \right)^2 \tag{4}$$

In the Formula (4):

- $A_i(g)$: array of intensity values with index i and glucose level g

- $V_j(A_i(g))$: average variance of the intensity series of $A_i(g)$ with starting index j

- $\underline{S}_j(A_i(g))$: a median of the intensity series of $A_i(g)$ with starting index j

- $I_j$: intensity value with index j

- l: the size of the intensity series

After calculating the average variance of each series of array, we chose a set number of the series with the highest average variance. Then, new array input data are defined by picking all indexes of the intensity array that are contained in the selected series of all samples. The new input data is the intensity array from which all intensity values whose indices are not selected have been removed.

**4.1.4. Implementation of the hotspot series extraction procedure.** We implement the hotspot series extraction procedure using Python. We set the size of the series as 10 values and the number of selected series in each sample is 500 series. With this dataset, the result is that 831 values is selected with some segments including index from 0 to 29, index from 31 to 542, index from 553 to 586, index from 629 to 647, index from 664 to 699, index from 703 to 743, index from 782 to 799, index from 992 to 1010, index from 1109 to 1122, index from 1171 to 1207, index from 1505 to 1523, index from 1620 to 1642, index from 1706 to 1715 and index from 1865 to 1883. In conclusion, the new input data has 831 values which contain 40.5% in comparison with the size of primitive data. In general, the advantage of hotspot series

extraction procedure is its simplicity in terms of implementation, low resource consumption, and short computation time, typically oscillating function O(n). The algorithm is also easily programmable in Python without requiring any external libraries. However, its drawback lies in its limited efficiency as it only reduces the length of the textual data without decreasing its dimensionality. Additionally, for Raman sequences with significant noise, the preprocessing algorithm may mistakenly identify noise-induced hotspot segments as characteristic labels of the data sample.

## 4.2. The fluorescence background removal method

To get rid of the fluorescence in the background of the original signal, several polynomial fitting algorithms were investigated. Zhao et al. [26] proposed the Vancouver Raman Algorithm (VRA), an iterative approach for correcting the baseline in Raman spectra. The VRA comprises two major steps: (1) noise reduction caused by the acquisition system or environmental interference, and (2) baseline correction or fluorescence removal.

- The VRA applies a smoothing window whose size is set by the user in the first phase. This manual technique allows the algorithm flexibility, but it may eliminate crucial information by combining Raman peaks or removing others. As a result, the outcome may be determined by the user's prior experience.

- In the second stage, VRA eliminates the most intense Raman peaks, and then an iterative method is used to adapt a polynomial function (usually of order five or six) to the smoothed Raman peaks.

Therefore, the Vancouver Raman Algorithm (IModPoly) approach suggested in the mentioned study was adopted to keep the data as clear as possible.The advantage of the VRA is its high efficiency, as it transforms the structure of the Raman sequence instead of reducing its size, making the characteristic labels easier to analyze and process by basic machine learning algorithms. Consequently, the impact of noise is reduced, and the effectiveness of the machine learning algorithm is less affected by noise. Conversely, the downside of the VRA is its complexity and difficulty in implementation, requiring the use of MATLAB, with a long computation time and high memory consumption. It is simple to proved in the experiment described in the next chapter. In order to get the pure Raman spectrum, one must first subtract the corrected polynomial from the raw Raman spectra and then add the resultant numbers.

In comparison to the hotspot series extraction procedure, the hotspot series extraction procedure removes Raman sequences that are deemed non-characteristic of the sample, while the VRA transforms the data sequence by trimming extreme points to smooth the Raman sequence. Because of the one-direction processing of these proposals, 3 duplications of root dataset is created and 2 of these duplications are pre-process by the hotspot series extraction procedure and the VRA. Disadvantagely, neither pre-processing method has the ability to restore the data to its original state because both involve operations that result in data loss.

## 5. Results and discussion

### 5.1. Experimental setups

We implemented our machine learning algorithm with three models, including Extra Trees, Random Forest and Support Vector Machine on a personal computer with adequate software and hardware configuration. Our computer has 16GB of RAM capability and an Apple M2 SoC model. Meanwhile, Python was selected to implement this algorithm because this programming language has a large number of libraries that strongly support machine learning

models. Specifically, we use MATLAB to apply fluorescence background subtraction algorithms and essential libraries in Python such as Scikit-learn to construct these machine learning models.

## 5.2. Experimental dataset

We apply the preprocessing procedure described in Sections 3.3 and 3.4 to our gathered dataset provided in Section 3.1. Nonetheless, because the distance between glucose levels of samples is not comparable, it is crucial to construct experimental datasets from the parent dataset. Therefore, the next step is to divide this experimental dataset into a train set and a test set. We subdivided the dataset based on k-fold validation to include data for training and testing.

## 5.3. Model hyper-parameter tuning using grid search

Finding a set of hyperparameters is always necessary. It is frequently necessary to look for a collection of hyperparameters that result in the best performance of a model on a dataset.

A search space is defined as part of an optimization technique. This may be seen as an n-dimensional volume, with each hyperparameter being a separate dimension and the scale of the dimension being the values that the hyperparameter can take on, such as real-valued, integer-valued, or categorical.

Grid search is excellent for testing combinations that are known to work well in general. Although it generally takes longer to complete, random search is excellent for discovery and obtaining hyperparameter combinations that would not have been predicted instinctively.

In particular, to get the best accuracy, the search space of our constructed models has several hyperparameters available for tuning as following:

1. Random Forest:

    a. n_estimators

    b. max_features

    c. max_depth

    d. max_leaf_nodes

    e. min_samples_split

    f. min_sample_leaf

2. Extra Trees:

    a. n_estimators

    b. max_features

    c. max_depth

    d. max_leaf_nodes

    e. min_samples_split

    f. min_sample_leaf

3. Support Vector Machine:

    a. C

    b. kernel

 c. gamma

 d. degree

After constructing the hyper-parameter search space of 75,200 possible fits, we use distributed training in parallel to speed up the training and validation phase. On instantiation we set the Pool's number of worker processes to the number of processors on the current system.

One shortcoming of grid search is that its dimensionality diminishes as the number of hyperparameters evaluated rises exponentially. However, there is no assurance that the search will provide the ideal answer, as it typically does so by aliasing around the correct set.

## 5.4. Separation of experimental dataset

To avoid overfitting, we implement k-fold cross validation, which is the process of using each subset as the test data set and the remaining subsets as the training data. It involves breaking a data set into k subsets. The performance metrics for each validation process are then averaged. There is not a single best indicator for evaluating machine learning algorithms because each approach has advantages and disadvantages. In the experiment, we divided the population equally into 5 subsets, with one subset used for the test set and the other four subsets used for the training set.

The test set for root dataset is comprised of 10 samples, each of which contains a distinct label-value from a collection of 10 label-codes. Currently, the train value data consists of forty samples, four of which have identical label-codes.

In addition, there are 450 iterations of training with each machine learning model including Extra Trees, Random Forest and SVM models.

## 5.5. Measurement metrics

The algorithm's efficiency is determined by the accuracy of the classification process with the data from the test set. Specifically, the algorithm with a specific machine learning model classifies a sample regardless of its label-code and defines a label-code for this sample. After that, the algorithm compares the predicted label-code to the existing label-code of this sample. There are possible 2 cases of this comparison:

- *True (T)*: The predicted label-code and existing label-code is the same.

- *False (F)*: The predicted label-code and existing label-code is different.

There are 4 metrics used to determine the effectiveness of our model. We use One-vs-rest (OvR) strategy to compute the Specificity (Sp), Sensitivity (Se) and ROC-AUC of our model:

The accuracy of our model is defined by Formula (5):

$$Acc = \frac{TP + TN}{TP + FP + TN + FN},\tag{5}$$

The model's Specificity (Sp) is defined as:

$$Sp = \frac{TN}{TN + FP}\tag{6}$$

The Sensitivity (Se) is defined as

$$Se = \frac{TP}{TP + FN},\tag{7}$$

where

- *TP*–Number of true positive instances

- *TN*–Number of true negative instances

- *FP*–Number of false positive instances

- *FN*–Number of falsenegative instances

Each machine learning model consists of $i = 450$ loops in which subsets are differently partitioned into train and test sets prior to training and accuracy values are determined after each turn. Afterwards, the mean metrics accumulated over 450 iterations are calculated. There are 3 ML model used in our scope of investigation: Extra Trees, Random Forest, and Support Vector Machine; thus, there are three values of average Accuracy for each model.

## 5.6. Experimental results

**5.6.1. Preprocessing techniques result comparison.**　This section discusses the outcome of applying our selected augmentation strategies for Raman spectroscopy to several machine learning models. The following are 3 scenarios for the dataset:

- Root dataset—without using any preprocessing method

- Hotspot series dataset–Using hotspot series extraction applied to the root dataset

- VRA dataset–Using fluorescence subtraction method to the root dataset

The Fig 4 describes the process of experiments with each of three datasets is input data of three different machine learning models. The kind of dataset which has highest accuracy score is use again in three machine learning models with hyper-parameter fine tuning.

The accuracy ratings of each machine learning model with three distinct input datasets are shown in Fig 5. The machine learning models include Extra Trees, Random Forest, and SVM, while the input datasets consist of root, hotspot series, and VRA. We utilise the accuracy score as the combined metric in the graphs as opposed to the three other metrics since this measure demonstrates clearly the effectiveness of machine learning in solving classification problems.

Fig 5 demonstrates that the VRA dataset whose root data is handled using the fluorescence subtraction approach enhances the performance of all machine learning models. Hence, the

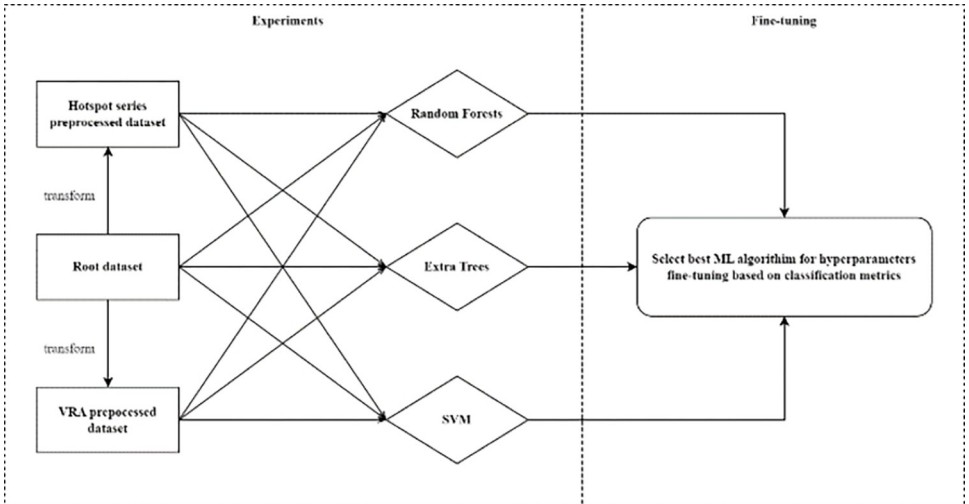

**Fig 4. Description of our experiment on generated dataset with ML models.**

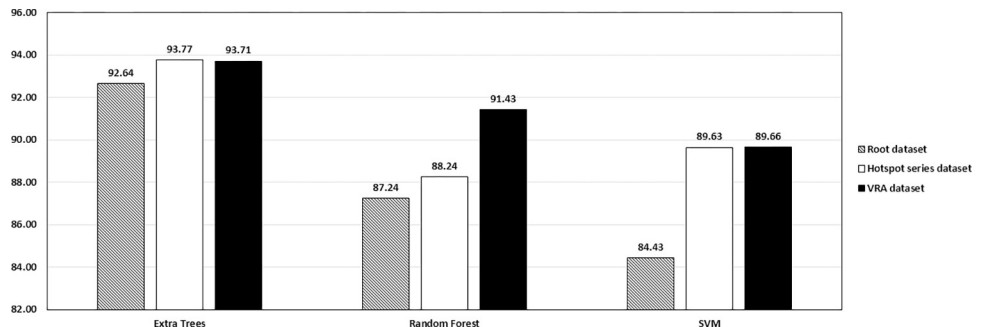

**Fig 5. Comparison between three machine learning models mean accuracy score among several Raman preprocessing algorithms.**

VRA dataset is used as the input dataset for all machine learning processes including fine-tuning. From these results, we conclude that the structure and characteristics of input data have a significant influence on the performance of machine learning models, and that two strategies for transforming input datasets demonstrate their efficiency by enhancing machine learning performance.

Root dataset—Without using hotspot series extraction.

Table 2 shows the result with each model in case of primitive data without any conversion. In each sample, the number of intensity Figs is 2048. The metrics are described in percentage unit (%):

The Extra Trees model has the highest accuracy score in this table, whereas the Random Forest and SVM models have lower Average Accuracy values. The accuracy ratings range from 84% to 92%, which is a moderately good range. Similar to the accuracy scores, the Extra Trees model has greater Specificity, Sensitivity, and ROC-AUC values than the other models. While the SVM model's estimate of its learning capacity is superior, its performance with this data is the worst. Yet, when comparing one component of One-vs.-Rest Specificity to another, we detect an anomalous characteristic. The SVM model has a better specificity than Random Forest, which is distinct from the three other measures. It indicates that the SVM model is more resistant to type I errors due to its unique processes for tackling classification issues.

*Preprocessed data—Using hotspot series extraction algorithm.*

Table 3 shows the result with each model in case of data after using hotspot series extraction. In each sample, the number of intensity Figs is only 831 which means its size is 40.5% of the size of the root dataset. The basic unit is also percent (%):

The Extra Trees model has the highest accuracy score in this table, whereas the Random Forest and SVM models have lower Average Accuracy values. The accuracy ratings range from 88% to 94%, which is a somewhat good range. Similar to the accuracy scores, the Extra Trees model has greater Specificity and ROC-AUC values than the other models. While the Random Forest has better Sensitivity than others.

**Table 2. Result of experiment with each model in case of root dataset.**

|  | Mean result over 450 iterations | | | |
|---|---|---|---|---|
|  | Acc | Sp | Se | ROC-AUC |
| Extra Trees | 92.64 | 99.19 | 92.73 | 99.23 |
| Random Forest | 87.24 | 87.47 | 85.40 | 98.59 |
| Support Vector Machine | 84.43 | 98.38 | 86.21 | 99.04 |

**Table 3. Result of experiment with each model in case of hotspot series preprocessed dataset.**

|  | Mean result over 450 iterations | | | |
|---|---|---|---|---|
|  | Acc | Sp | Se | ROC-AUC |
| Extra Trees | 93.77 | 99.30 | 93.67 | 99.48 |
| Random Forest | 88.24 | 98.70 | 99.30 | 99.02 |
| Support Vector Machine | 89.63 | 98.84 | 98.60 | 99.37 |

*Preprocessed data—Using fluorescence subtraction method. VRA preprocessed dataset*

These models' optimal hyperparameters were determined via grid search. A set of parameters were initialized and are being used as the model's parameters space for training sequentially over Extra Trees, Random Forest and SVM classifiers. We have also conducted trials utilizing the data produced by fluorescence subtraction algorithms. Using VRA improves the accuracy of every model in a dataset, as seen in the table below. Therefore, this approach may be used as an effective preprocessing algorithm for high-dimensional spectral data to solve the problem.

Our observations on the metrics indicates that we had produced an increase in accuracy scores in every model as a result of our preprocessing recommendations. The final result of VRA preprocessing technique is more prominent. Thereby, the differences were not statistically significant among both scenarios.

According to Table 4, the accuracy scores for the performance of three machine learning models handling the VRA dataset are somewhat higher than the accuracy scores for the root dataset. Particularly, the percentages vary from 90 to 94 percent, which is quite high. Apparently, the noise-elimination method has a substantial effect on the activity of machine learning models used to tackle this classification issue. In conclusion, the fluorescence subtraction approach provides the greatest efficiency among the three machine learning models for addressing the categorisation of Raman spectroscopy samples. Based on the experiment, we will use the VRA dataset as the input dataset for machine learning models in subsequent experiments including a process of fine-tuning.

**5.6.2. Machine learning model fine-tuning process.** *Using grid search as a hyper-parameter tuning method.* As a result, we continue to spend the efforts to deeply investigate the dataset and have constructed the grid search space for models, which consists of 75,200 possible hyperparameters. Using VRA as a fluorescence subtraction approach may thereby increase the accuracy of these models. Consequently, a grid search process is carried out to find the optimal parameters for our models. The results are described in Table 5. This table also compares the performance of three models between fine-tuned and random parameters:

By utilizing the grid search technique, the accuracy score in every model is significantly enhanced. Eventually, the best averaged model that can be acquired is Extra Trees using

**Table 4. Comparison of model mean accuracy for the effectiveness of the VRA—fluorescence subtraction algorithm over 450 iterations.**

|  | Extra Trees | Random Forest | Support Vector Machine |
|---|---|---|---|
| **VRA Preprocessed** | **93.71** | 91.43 | 89.66 |
| Root dataset | 92.64 | 87.24 | 84.43 |
| **Improvement in Accuracy** | **+1.07** | **+4.19** | **+5.23** |

**Table 5. Comparison of model accuracy for effectiveness of Grid search for hyperparameter tuning.** The results are measured by using VRA as a preprocessed method for input data.

|  | Extra Trees | Random Forest | Support Vector Machine |
|---|---|---|---|
| Fine-tuned by Grid search | **97.27** | 95.61 | 97.04 |
| Random parameters | 93.71 | 91.43 | 89.66 |
| **Improvement in Accuracy** | **+3.56** | **+4.18** | **+4.38** |

preprocessed data and a grid search process. Table 6 shows the accuracy, specificity, sensitivity, and ROC-AUC of the best model applied to VRA after hyperparameter tuning.

## 6. Conclusion

Based on studies into the correlation between a person's glucose level and Raman spectroscopy reflected from various bodily areas, we suggested a method for assessing glucose utilizing a Raman spectrometer and a machine learning system with many models. Before the machine learning models anticipate the glucose level from the samples, the spectrometer generates Raman spectra from human samples. We utilized the Technospex uRaman—Ci spectrometer to build a dataset from glucose-mixed fluids with varying glucose concentrations. Before being labeled as primitive data, the dataset has been cleansed of noise. We also utilized feature-extraction techniques to enhance the performance of machine learning systems. Fluorescence subtraction is utilized to extract features from primitive data. We also developed a novel extraction method based on retaining data whose indices belong to the series with the biggest volatility. The input data for the machine learning model could be basic data or data extracted during the extraction procedure.

Before testing the accuracy of the trained classification model of samples, we designed experiments in which machine learning algorithms extracted characteristics from the dataset.

Experiments utilized three models: Extra Trees, Random Forest, and SVM model. The results proved the efficiency of machine learning on classification problems as well as the pre-processing procedures that meet our requirements. The accuracy ranges from 80% to 97%, while the extraction process increases the accuracy of each machine learning model in the same experimental dataset. However, our research have several obstacles such as pure-glucose liquid samples not being similar to human biological samples in practical applications, as these samples are subject to various related substances that affect the quality and representation of glucose characteristics in the obtained Raman spectrum. Additionally, the number of samples and labels is limited, and the differences in glucose concentration between the labels are quite sparse.

From these drawback, we will raise the dataset complexity in the future by increasing the number of labels and the number of samples for each label. Non-invasive sampling methods for Raman spectroscopy data collection will also be thoroughly investigated, and we will continue to improve the machine learning models and preprocessing techniques used to extract characteristics from the dataset.

**Table 6. Our best model's metrics of Grid search after hyperparameter tuning, with VRA applied.**

|  | Mean result over 450 iterations | | | |
|---|---|---|---|---|
|  | Acc | Sp | Se | ROC-AUC |
| Extra Trees | **97.27** | **99.10** | **96.45** | **99.86** |
| Random Forest | 95.61 | 97.81 | 94.63 | 99.54 |
| SVM | 97.04 | 98.82 | 93.16 | 99.70 |

## Author Contributions

**Formal analysis:** Tri Ngo Quang, Duc Le Anh.

**Investigation:** Doanh Sai Cong.

**Methodology:** Tri Ngo Quang, Huong Pham Thi Viet.

**Project administration:** Tung Nguyen Thanh.

**Supervision:** Tung Nguyen Thanh.

**Validation:** Duc Le Anh.

**Writing – original draft:** Tung Nguyen Thanh.

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
