## [Decision Letter · Decision Letter 0]

15 Jul 2024

PONE-D-23-21128Classification of glucose-level in deionized water using machine learning models and data pre-processing techniquePLOS ONE

Dear Dr. nguyen thanh,

Thank you for submitting your manuscript to PLOS ONE. After careful consideration, we feel that it has merit but does not fully meet PLOS ONE’s publication criteria as it currently stands. Therefore, we invite you to submit a revised version of the manuscript that addresses the points raised during the review process.

We look forward to receiving your revised manuscript.

Kind regards,

Hongju He

Academic Editor

PLOS ONE

“Funding: Research grants from Viet Nam National University, Hanoi and research support (including salaries, equipment, supplies, reimbursement for attending symposia, and other expenses) by organizations that may gain or lose financially through publication of this manuscript.”

5. We note that you have indicated that there are restrictions to data sharing for this study. PLOS only allows data to be available upon request if there are legal or ethical restrictions on sharing data publicly. For more information on unacceptable data access restrictions, please see http://journals.plos.org/plosone/s/data-availability#loc-unacceptable-data-access-restrictions.

7. We notice that your supplementary figures are included in the manuscript file. Please remove them and upload them with the file type 'Supporting Information'. Please ensure that each Supporting Information file has a legend listed in the manuscript after the references list.

Reviewers' comments:

Reviewer's Responses to Questions

**Comments to the Author**

1. Is the manuscript technically sound, and do the data support the conclusions?

Reviewer #1: No

Reviewer #2: Yes

Reviewer #3: Partly

2. Has the statistical analysis been performed appropriately and rigorously? 

Reviewer #1: No

Reviewer #2: Yes

Reviewer #3: No

3. Have the authors made all data underlying the findings in their manuscript fully available?

Reviewer #1: No

Reviewer #2: No

Reviewer #3: Yes

4. Is the manuscript presented in an intelligible fashion and written in standard English?

Reviewer #1: No

Reviewer #2: No

Reviewer #3: Yes

5. Review Comments to the Author

Reviewer #1: This paper titled "Classification of glucose-level in deionized water using machine learning models and data pre-processing technique" is lack of innovation and badly written. The structure of this article is chaotic, far from the requirements of this journal. Therefore, I suggest the rejection of this paper.

Reviewer #2: The authors described the classification of glucose-level in deionized water using machine learning models coupled with Raman spectroscopy. Before publication can be considered, the authors need to address the following issues:

1. Authors should specify and comment (in abstract) on their main findings

2. The authors indicated measuring the analyte at 10 different concentrations but do not mention how many times they measure each concentration, especially the best peak.

3. How about signal reproducibility of the proposed method?

4. The authors should highlight the limitations and strengths of their proposed method

5. Change Fig 1a to Fig 2a and Fig 1b to Fig 2b

6. The authors used lots of abbreviations that should be properly defined in the text

Reviewer #3: 1. No reference in section 1

2. Comparison with the state-of-the-art is missing

3. Future direction, limitations and threats to validity is missing

4. Data collection part is very impressive , congratulations

5. Last and the most major concern is the lack of novelty in the work, only already available methods are implemented in the work . Please keep in mind that in current state the work cannot be published, you have to make significant improvements in the methodology and experimentation required for the publication in this highly reputed SCI indexed journal. I hope you understand my concern and I expect you will overcome this issue by incorporating the novelty in the methodology.

Best Regards

6. PLOS authors have the option to publish the peer review history of their article (what does this mean?). If published, this will include your full peer review and any attached files.

Reviewer #1: No

Reviewer #2: No

Reviewer #3: No

---

## [Author Response · Author response to Decision Letter 0]

23 Aug 2024

ANSWER FILE

Reviewer 1: 

Classification of glucose-level in deionized water using machine learning models and data pre-processing technique

The overall quality of the paper is poor. This manuscript was not well written and suggest a rejection of the manuscript. The line in the whole document is not numbered

1. I suggest the authors get editing help from someone with full professional proficiency in English. There are improper or ambiguous English usages. The writing of the paper needs to be polished.

2. Abstract is not well written.

The abstract is rewritten with better structure.

3. The introduction section is not well organized.

The introduction is rewritten with better structure

4. Methodology is poorly written.

We restructure the methodology section and divide into 3 sections including Related Work, Related works and description of methodology, Introduction to the dataset, Data preprocessing methods in proposal. Each section focuses on a specific tasks related to our proposal.

Reviewer 2:

The authors described the classification of glucose-level in deionized water using machine learning models coupled with Raman spectroscopy. Before publication can be considered, the authors need to address the following issues:

1. Authors should specify and comment (in abstract) on their main findings.

The abstract is rewritten with better structure. We specify the results about improvement of machine learning.

2. The authors indicated measuring the analyte at 10 different concentrations but do not mention how many times they measure each concentration, especially the best peak.

We collected 5 Raman sequences for each glucose fluid with specific concentration. With 10 different concentrations, we collected 50 sequences.

3. How about signal reproducibility of the proposed method?

Both the hotspot series extraction procedure and the VRA have the ability to restore the data to its original state because both involve operations that result in data loss. The hotspot series extraction procedure removes Raman sequences that are deemed non-characteristic of the sample, while the VRA transforms the data sequence by trimming extreme points to smooth the Raman sequence. This argument is added in part 3.2 of rewritten paper.

4. The authors should highlight the limitations and strengths of their proposed method

We add the advantage and disadvantage of our 2 proposals for pre-processing data including hotspot series extraction procedure and Vancouver Raman Algorithm in Part 3.1.4 and 3.2. In categories of machine learning, we use existing algorithms without modification so we focus on how these algorithms handle Raman data instead of their cons and pros.

5. Change Fig 1a to Fig 2a and Fig 1b to Fig 2b

We changed Fig1a to Fig 2a and Fig 1b to Fig 2b.

6. The authors used lots of abbreviations that should be properly defined in the text

We rewrite clearly an abbreviation ML by full word “Machine Learning”, with abbreviation VGA, we correct it to VRA as explanation of Vancouver Raman Algorithm.

Reviewer 3:

1. No reference in section 1

The section 1 is introduction, in this section, we concentrate the general ideal and process of implementation for our research so the number of research is limited. In rewritten introduction, several references about the problem defining are added.

2. Comparison with the state-of-the-art is missing.

We have several comparisons with the state-of-the-art technologies such as Convolutional Neural Networks (CNNs) or Long Short-Term Memory (LSTM) networks. We define the advantage and disadvantage of these deep learning model and, thus, explain the reason to not use this model. This argument is added in section 2.3 of rewritten paper.

3. Future direction, limitations and threats to validity is missing

We add drawbacks of our research as well as the future development in Section 5: Conclusion

4. Data collection part is very impressive, congratulations

5. Last and the most major concern is the lack of novelty in the work, only already available methods are implemented in the work. Please keep in mind that in current state the work cannot be published, you have to make significant improvements in the methodology and experimentation required for the publication in this highly reputed SCI indexed journal. I hope you understand my concern and I expect you will overcome this issue by incorporating the novelty in the methodology.

We define that the novelty of our research is creation of dataset by unique research strategy as well as the analyzation of our dataset for proposing new data pre-processing method. We describe it in section 2.3.

---

## [Decision Letter · Decision Letter 1]

10 Sep 2024

Classification of glucose-level in deionized water using machine learning models and data pre-processing technique

PONE-D-23-21128R1

Dear Dr. nguyen thanh,

We’re pleased to inform you that your manuscript has been judged scientifically suitable for publication and will be formally accepted for publication once it meets all outstanding technical requirements.

Kind regards,

Hongju He

Academic Editor

PLOS ONE

Additional Editor Comments (optional):

Reviewers' comments:

Reviewer's Responses to Questions

**Comments to the Author**

1. If the authors have adequately addressed your comments raised in a previous round of review and you feel that this manuscript is now acceptable for publication, you may indicate that here to bypass the “Comments to the Author” section, enter your conflict of interest statement in the “Confidential to Editor” section, and submit your "Accept" recommendation.

Reviewer #2: All comments have been addressed

Reviewer #3: All comments have been addressed

2. Is the manuscript technically sound, and do the data support the conclusions?

Reviewer #2: Yes

Reviewer #3: Partly

3. Has the statistical analysis been performed appropriately and rigorously? 

Reviewer #2: Yes

Reviewer #3: I Don't Know

4. Have the authors made all data underlying the findings in their manuscript fully available?

Reviewer #2: No

Reviewer #3: Yes

5. Is the manuscript presented in an intelligible fashion and written in standard English?

Reviewer #2: Yes

Reviewer #3: Yes

6. Review Comments to the Author

Reviewer #2: (No Response)

Reviewer #3: Almost all major issues have been resolved hence I think that this revised version can be accepted now .

7. PLOS authors have the option to publish the peer review history of their article (what does this mean?). If published, this will include your full peer review and any attached files.

Reviewer #2: No

Reviewer #3: No

---

## [Editor Report · Acceptance letter]

8 Oct 2024

PONE-D-23-21128R1 

PLOS ONE

Dear Dr. nguyen thanh, 

I'm pleased to inform you that your manuscript has been deemed suitable for publication in PLOS ONE. Congratulations! Your manuscript is now being handed over to our production team.

Kind regards, 

on behalf of

Dr. Hongju He 

Academic Editor

PLOS ONE